# Intergenerational Narrative Learning to Bridge the Generation Gap in Humanistic Care Nursing Education

**DOI:** 10.3390/healthcare9101291

**Published:** 2021-09-29

**Authors:** Yu-Lun Kuo, Jian-Tao Lee, Mei-Yu Yeh

**Affiliations:** 1Department of Nursing, Tzu Chi University of Science and Technology, Hualien 970, Taiwan; ss246@ems.tcust.edu.tw; 2School of Nursing, College of Medicine, Chang Gung University, Taoyuan 333, Taiwan; jtlee@gap.cgu.edu.tw; 3Nursing Department, Chang Gung Memorial Hospital, Taoyuan 333, Taiwan; 4College of Health and Nursing, Mei Ho University, Pingtung 912, Taiwan

**Keywords:** humanities, intergenerational relations, narration, learning, qualitative analysis

## Abstract

Background: The development of nursing students’ ability to practice humanistic care is extremely important. Methods: This study explored students’ learning experience when providing humanistic care for older adults with chronic diseases while employing intergenerational narrative learning. An exploratory descriptive qualitative study design was adopted. Results: We analyzed evaluations from 35 students who completed the course, in which intergenerational narrative learning was employed. Evaluations contained open-ended questions that asked students to reflect upon their experiences and describe their perceptions, thoughts, and feelings after the course. Three main themes were revealed by thematic analysis: direct interaction supersedes knowledge in books, the framework for improving humanistic caring, and internalization of the importance of humanistic care in nursing. Conclusion: An awareness of patients’ perspectives inspired the students in their development toward a more profound caring attitude. The intergenerational narrative learning teaching strategy could foster professional and humanistic-centered care in nursing students.

## 1. Introduction

The development of nursing students’ ability to provide humanistic care to patients with chronic diseases is extremely important. In addition to the learning of professional knowledge and skills, nursing education should also focus on the cultivation of humanistic caring. Humanistic care refers to nurses providing patients with physical, psychological, and psychospiritual care and support, to promote patients’ overall health [1]. The literature shows that the humanistic caring ability of undergraduate nursing students can be effectively enhanced through a narrative approach [2].

The narrative approach plays an important role in cross-disciplinary learning, including education, healthcare, and humanities [3,4,5]. The narrative approach allows for a rich description of patients’ experiences and reflection on meaning [6]. It helps nursing students to understand patients and illuminate the intricacy of specific phenomena and the paradigms that shape peoples’ experiences [7]. Therefore, the storytelling process is a fundamental element in the narrative approach, as it provides the opportunity for reflective dialogue [8], which allows for the capture and transfer of tacit knowledge [9,10]. The philosophy underpinning the narrative approach is dependent on accessing real people, in real settings, wherein patients paint their experiences in words. It leads to an increase in the depth of understanding of these individuals [11].

The narrative approach is useful for nursing students to understand the patients’ disease experience, as they discover themselves and their own humanity enhanced through patients’ stories [12,13]. Through understanding patients’ perspectives, nursing professionals learn to respect the emotional experience and exercise the empathetic side of human nature when caring for patients suffering from disease [14,15]. Each generation has different memories, language, habits, beliefs, and life lessons affected by diverse social background factors [16]. In addition, nursing students generally lack experience interacting with patients or older adults experiencing clinical health issues, which produces a sense of alienation, exclusion, and the perception of a generation gap [17].

In view of this, it is necessary to use a generational perspective that allows us to examine shared experiences and similarities within generations, to promote insight and awareness about generational diversity [18]. The narrative approach can be applied through intergenerational learning based on Dewey’s (1938) theory of the three-dimensional space narrative structural approach that leads to meaning. This includes (1) interaction, (2) continuity, and (3) situation [19]. Intergenerational learning has been shown to improve practical understanding, reduce stereotyping, and alleviate tensions between generations, and encourages nursing students to integrate previous learning and experience for improved nursing care [20,21].

Taiwan is facing the challenge of an aging population, and the concomitant complexities of chronic disease. It is necessary to promote awareness and insight among nursing students to understand the needs of patients and older adults with chronic diseases (OACD). Therefore, the purpose of this study was to explore nursing students’ learning experience when employing intergenerational narrative learning (INL) focused on the humanistic care of OACD. To the best of our knowledge, this study was the first to apply INL as a teaching strategy and qualitatively analyze the effect within a professional nursing curriculum.

## 2. Methods

### 2.1. Design

A qualitative study design was used to explore and understand students’ INL experiences. An open-ended qualitative exploration questionnaire was used to generate and guide the collection of data for the study. Participants were asked to describe their INL experiences, including reflections and personal thoughts based on the narrative approach they engaged in with older patients who were chronically ill. Thematic analysis was used to explore the INL experience of students in the older patients’ chronic disease-care course.

### 2.2. Participants

Purposive sampling of nursing students in a university in eastern Taiwan was employed. The inclusion criteria were (1) no previous professional work-related nursing experience; (2) third-year full-time undergraduate nursing student; (3) aged not less than 20 years old. Exclusion criteria were (1) interdisciplinary students and (2) foreign exchange students. In total, 35 full-time undergraduate third-year nursing students (33 female, 2 male) were recruited into the study (Table 1)

### 2.3. Intergenerational Narrative Learning

Intergenerational narrative learning was used as a teaching strategy in a course entitled “Nursing Care and Management of Chronic Disease” to bridge the generation gap between students and OACD. The course was an 18-week course in the 3rd year of the nursing program. The INL course lasted for was five weeks long and met for two hours each week. The concept of holistic care for chronic disease sufferers was introduced in the first week. In the second week, students were instructed on methods of inquiry to understand the impact of chronic disease on physical and mental health among older patients. In the third and fourth weeks, the INL was implemented, with five older patients suffering chronic disease invited to attend the lessons. At the end of each session during the third and fourth weeks, students were required to submit within one week, in written form, their reflections of the narrative approach. In the fifth week, students were asked to make an oral presentation of their learning experiences from INL to the class and the patients.

The INL was conducted in three steps: (1) The instructor guided the INL activity and provided interview guidelines for the nursing students and patients. The intergenerational interview guidelines were as follows: (A) Would you please describe your experiences with your illness, such as symptoms, seeking medical attention, and the treatment process? (B) Please tell us in your own words how you experience your illness. (C) How has the disease affected your life? (D) Could you please tell us your experiences of the difference in your life, pre- and post-disease diagnosis? (E) Could you tell us, post-diagnosis, what aspect of your illness caused you the greatest worry? (2) Five groups were formed for intergenerational dialogue, with each group consisting of one patient and seven students. The life stories, as told by the patients, would allow nursing students to explore and understand the experiences of chronic disease on a directly intimate basis; (3) students participated in a dialogue with representatives of each group for further sharing about what they had learned from INL. In this reflective practice within INL, students systematically thought about their feedback and responses to improve future nursing actions and responses.

### 2.4. Data Collection

This study was conducted from March to June 2018. The students were informed of the purpose and process of the INL course. At the end of the course, students wrote their reflections and knowledge gained and shared personal thoughts regarding the learning experiences. Data were collected within one week after the completion of the course. The open-ended questions were as follows: (1) What is your impression or opinion of people with chronic diseases? (2) What have you learned from the experiences with disease shared by the older patients? (3) How do you think the course may affect your nursing care? Each question required a minimum 200-word response from the student participants. The data in the reflective essay were accessed, collated, and then subjected to thematic analysis.

### 2.5. Analysis

For this qualitative study, thematic analysis was used to analyze the data. The study by Gordon et al. [3] suggested preliminary sorting of data during an initial reading of the nursing student reflections by the research team. A second reading allowed the researchers to encode the first meaningful text data and create codes. The third step focused on defining common themes. The fourth step classified themes by similarity and labeled definitive themes. Finally, the overall structure was refined, as the text data was reviewed repeatedly to increase the depth and breadth of each category.

### 2.6. Rigor

To ensure rigor, the study adhered to four major criteria: true value, applicability, consistency, and neutrality [22]. To ensure true value, findings were verified with participants (member check) and discussed findings with the other authors (peer debriefing) when points of view differed during data interpretation and analysis. Consistency was enhanced by closely following coding book guidelines to explain reasons for including and excluding data. To ensure neutrality, assumptions and biases of participants’ learning experiences were recorded.

### 2.7. Ethical Consideration

The study protocol was approved by the Taiwan National University Institutional Review Board (IRB Number 201708ES010). The nursing students involved in the study were informed of the research purpose and process in detail. Informed consent was obtained prior to their participation in the study. The nursing students, while being unable to withdraw from the course itself, retained the right to refuse to fill out the class reflection reports without penalty. All data were kept confidential in accordance with the rights to privacy laws and ethical standards.

## 3. Results

The study findings resulted in the deconstruction of the INL impact on students and the design of the framework for improving humanistic caring through INL (Figure 1). The overarching theme was “awareness through learning”, and three main categories were identified: (1) direct interaction supersedes knowledge in books, (2) framework for improving humanistic caring, and (3) internalization of the importance of humanistic care in nursing. Our result showed that the perspectives of nursing students could be altered when actively engaged in understanding the life stories of older people suffering from chronic disease. This awareness of the patient perspective inspired the students in their development toward a more profound caring attitude.

### 3.1. Direct Interaction Supersedes Knowledge in Books

#### 3.1.1. Breaking through Stereotypes

Nursing students can enhance their professional knowledge through self-directed education based on availability and personal interests. However, the unique life trials of chronically ill patients are more suitably learned through the narrative approach. The transition from fighting disease to peaceful coexistence with the disease is inspiring, which others can empathize with. Through INL, students were able to vicariously experience the reactions and adjustments of the patients dealing with different stages of their diseases. The students adjusted their presumptions of chronically ill patients’ experiences.


*“In my memory, older people rarely maintain an optimistic outlook when dealing with chronic illness. Generally, we may blame fate or God while feeling uncomfortable, but when [the older man] felt pain, he tried to talk to his cells to relieve the pain. He is very different [from most older people].”*
(Group A-5)

#### 3.1.2. Deeper Awareness from Narrative Inquiry

Through course work, students were expected to develop a deeper comprehensive understanding of caring for patients with chronic diseases. The dilemmas and problems faced by patients throughout their disease experience cannot be effectively presented and understood without direct human interaction. Through INL, students were able to understand, on a more personal level, the process of dealing with chronic disease from the perspective of an older chronic disease sufferer. The acquired knowledge exceeded the level of book knowledge.


*“We can understand more about the problems and difficulties in cases actually encountered, which is much closer to reality than those mentioned in the textbook.”*
(Group D-6)


*“I know more about their moods and how to deal with chronic diseases and the process of their mood changes. Those are things we can’t learn from the textbook.”*
(Group D-2)

### 3.2. Framework for Improving Humanistic Caring

#### 3.2.1. Self-Awareness

By witnessing the attitudes of patients dealing with chronic diseases, students were capable of a greater appreciation for the dignity, value of life, and improving humanistic caring. This initiated self-reflection in the students, and they reconsidered their attitudes toward life’s difficulties. Students also increased self-awareness of the current situations and accepted life’s challenges as lessons, rather than complaining about difficulties.


*“He is so active and cheerful despite his chronic illness. What reasons do young people have to complain about their current lives?”*
(Group C-6)


*“Seeing [patients] so calm and peaceful, I learned to take life’s difficulties as challenges.”*
(Group E-5)

The positive attitude of the older patients in the face of chronic disease inspired students to realize that life does not always go smoothly. Students learned from the patients to be brave and open-minded in facing the challenges of different stages of life.


*“The older people often experience serious pain due to illness, but they are always positive and just tell us to be peaceful and enjoy life in order to ease pain. This is what we need to learn; whether it is an illness or doing things we don’t want to face, we must be like these elders and remain positive all the time.”*
(Group A-2)


*“We have learned that when we face difficult things, we must learn from the attitudes of these elders to be brave and open-minded.”*
(Group D-3)

#### 3.2.2. Live in the Present Moment

For young students, there is a chasm of difference in life experience while trying to understand the emotions of an older person suffering from chronic disease. However, through intergenerational dialogue, students witnessed the impermanence of life and the nature of being old and sick and were forced to accept that life continues despite difficulties. The meaning of “cherish the present moment” is a difficult concept for many to understand. From the experiences of the patients, this might be understood as unique to the individual as their experience of their disease condition.


*“People will face birth and death; it is a necessary part of life, so we must cherish the present moment.”*
(Group A-1)


*“The dialogue with the older people reminded me of the importance of being healthy and cherishing my family and the present moment.”*
(Group B-4)

### 3.3. Internalization of the Importance of Humanistic Care in Nursing

Intergenerational narrative learning was able to inspire nursing students to demonstrate the attitude and characteristics of humanistic caring. Students understood the importance of empathy and tolerance and learned to respect individual differences while shouldering the responsibility and mission of nursing staff to implement humanistic care. This category contains four themes: (1) deeper empathy and greater tolerance, (2) respect of differences in individual needs, (3) development of holistic empathetic humanistic nursing care, and (4) professional nursing responsibility.

#### 3.3.1. *Deeper Empathy and Greater Tolerance*

Through active listening to the life stories of patients with chronic diseases and interacting with the patients, students were able to become immersed in the patients’ experiences. As the emotions of the personal stories changed over time, the mood of the students paralleled the shifts between despair and hope. Students vicariously experienced the hardship and frustration of patients with chronic diseases during the course of the disease.


*“During the discussion, I gradually understood their background story. When they described the stages of their illness, I could feel their emotional ups and downs, from shock to depression, and then relief.”*
(Group B-4)

This emotional comprehension allowed students to fully immerse themselves into the emotional experience of the older patients and also to expand this awareness to their own families and apply it to other clinical patients.


*“My mother just finished a gynecologic tumor resection last year. At that time, she was very anxious, but I did not have empathy for her, and just said not to worry about it. But, after listening to the anxiety and uneasy moods shared by the breast cancer patients, I think [my mother] must have felt extremely uncomfortable at the time.”*
(Group B-6)


*“After listening to the older people experiences with illness and treatment, I realized that pain and drug side effects are not something that ordinary people can understand. I think I can have more empathy for patients after seeing their sadness and crying during the [discussion].”*
(Group B-1)

#### 3.3.2. Respect of Differences in Individual Needs

The construction of professional nursing knowledge may, at times, create a barrier between nursing staff and patients. Professionals’ subjective cognition, coupled with decisive intervention in the treatment of diseases, may inadvertently fail to address the other real needs of patients with chronic diseases. Through interactive dialogue and active listening to the older chronic disease sufferers, students realized that each patient’s situation was unique, and their interpretation of the disease and care needs was correspondingly different.


*“After the sharing of older people, I better understood that the needs of each person are different; I can’t measure others by my own perspective. For example, if a patient needs to sleep, instead of just giving him a soft and comfortable mattress, we need to consider that he may need to raise his foot or head, which may be based on what he really needs.”*
(Group A-1)

Actively empathizing with patients created a communication bridge and potential for consensus on care between nursing staff and patients. This could allow for greater efficacy in meeting the real clinical and emotional needs of patients.


*“A caregiver needs to pay more attention to experience, observe each patient’s real needs, and decide which methods are suitable to resolve them.”*
(Group E-4)


*“Although we can’t really feel the pain of patients with chronic diseases, through communication we can understand what they need.”*
(Group A-5)

#### 3.3.3. Development of Holistic Empathetic Humanistic Nursing Care

Throughout the course, students were directed to consider the meaning and value of nursing care. Learning to take on the perspective of the disease sufferer and the patient-specific needs was core to empathetic humanistic nursing. The focus of care was the disease in its’ superficial appearance but also the multiple expressions and impact it might have.


*“Quality of life is closely related to health. We should not only look at the surface effects of disease, but go deeper into the patients’ psychology, and explore the effects that affect their quality of life from a multi-faceted perspective. … After intergenerational learning, we have gained a lot of valuable knowledge and experience!”*
(Group D-2)

Beyond traditional nursing practices, students came to understand that patient-centered care covers multiple aspects of the human experience in body and mind.


*“Through the sharing of the older people about their experiences with chronic disease, it makes me understand the importance of listening to and accompanying patients dealing with the disease, and supporting them in body and mind.”*
(Group D-5)

The nursing students learned the importance of providing overall medical care, as well as psychological and emotional support during patients’ potentially most difficult life experiences.


*“We hope that we can become a part of a support system for patients, and accompany them through their most difficult time. … In addition to implementing the measures according to doctors’ advice, we also need to provide psycho-spiritual support.”*
(Group E-7)

#### 3.3.4. Professional Nursing Responsibility

The intergenerational dialogue was able to connect students with their nursing mission and aspirations in a more profound manner than traditional classroom teaching. Based on the patients’ needs, nursing students could understand their own purpose within the nursing profession, their responsibilities, and commitments to care. Nursing students are expected to become professional nursing staff, capable of alleviating patient suffering through pharmaceutical pain relief, as well as through human connection. The co-mingling of clinical care with human connection is the essence of humanistic nursing care.


*“After patients are diagnosed with a chronic illness, we must try our best to get them back to the healthiest condition possible, giving them individual care and support based on their unique requirements.”*
(Group D-4)

Students also recognized that they care for their patients, but they also care about a mission for the socially disadvantaged.


*“After the course, I was thinking about people who are poor and sick, and how we can improve their quality of life and really help them.”*
(Group E-3)

During the learning process, students expanded their understanding of their impact on chronic disease care. They became more aware that they must personally actively pursue disease prevention based on the trends of chronic diseases.


*“We are aware of the importance of health promotion and self-care ability due to the increase of patients with chronic illnesses. We should focus not only on hospitals and institutions but also on communities; this is also the region we need to promote awareness of in the future.”*
(Group A-7)

## 4. Discussion

Nursing care is a dynamic process of personal interaction, taking into account social and physical environments [6]. This study found that the OACD and students interacted well, despite the differences in social backgrounds, age, and cultural contexts. When the patients described their experiences with illness, their facial expressions and body language communicated on yet another level of understanding. The results of this study support the findings by Haydon et al. [23], which showed that the narrative approach can deeply explore the patient care process from the patient’s perspective. This result confirms that the narrative approach is not simply storytelling; it is a method of inquiry that uses storytelling to delineate nuance [6]. The narrative approach improves people’s understanding of others’ experiences and enhances the level of understanding by continuous interaction with others, thus offering fresh opportunities for respectful, empathetic, and psychospiritual healthcare [24].

Similarly, the OACD’s positive attitudes toward life, response styles, and behaviors during treatment made a strong impression on students. This inspired student introspection, which altered the student’s perspectives, and will affect disease interpretations of patients in the future. The process of INL offers nurses rich insights for understanding the real experiences of older patients with chronic illness [25]. Students’ beliefs and life values were influenced. Students became more aware of the importance of mindfulness, to cherish the present moment despite any difficulties they were currently facing. The results of this study corroborate that due to the narrative process, students were better able to access their practical knowledge to engage in nursing care practice. The narrative approach improves professional identity and develops an ethic of caring through reflection [9].

The study by Augustin and Freshman [26] emphasized that education is about students acquiring knowledge from instructors, as well as gaining knowledge from their accumulated personal involvement and experience. The narrative approach provides an opportunity to come to a better understanding of what it means to be human and what health means to people [27]. Through the experience narrative of the OACD, students were guided to explore the details of the effects of chronic disease from the patients’ perspective [6]. Only when the patients’ needs are truly understood can nurses better communicate with the patients and avoid conclusions based on their own subjective perceptions [25]. INL also simultaneously guided students toward improved self-awareness and personal growth.

This study found that INL effectively bridged the generation gap. The research showed that students could improve their empathetic understanding of patients through direct interaction [26]. Students enter the world of the OACD and “walk a mile in their shoes” to understand their lived experiences. Through empathetic understanding, communication between the generations improved, which generated more positive attitudes toward older patients from the students [28]. Nursing students were able to more readily accept the perspectives and experiences of the OACD, which also acted to promote changes in beliefs, attitudes, and understanding. The experience sharing provided students the opportunity to learn from OACD who actually went to the classroom and also gain practical real-world experience, which remains closely integrated with the curriculum.

Based on the results of this study, suggestions are as follows: (1) To enhance humanistic nursing care for nursing students, the opportunity to accumulate experience through interaction and participating with others is an indispensable learning activity in nursing education. In the future, INL can be integrated into nursing courses, allowing students to actually interact with the patients, practice communication, provide care and enlighten sensibility while integrating with subsequent practice courses to improve clinical adaptability; (2) the implementation method of INL is concrete and feasible and is readily integrated within the curriculum design. In the future, it will continue to be discussed in depth academically, and it will be developed into a set of teaching models, and quantitative research methods will be used to verify the effectiveness; (3) recruit patients for INL to establish teaching resources as living materials for courses to enhance the diversified aspects of learning.

Several limitations existed in this study. First, the intergenerational narrative learning course was offered for only five weeks with a single meeting each week. This time frame was extremely limiting and should be extended over several months. Second, the number of interviewees with chronic diseases was limited. Only five people were willing to participate. A study with a greater variety of patients with chronic disease should be employed in future studies. Third, the collected intergenerational narrative learning data relied solely on the written texts of the nursing students. The richness and depth of text content might be affected by the emotional fluency of students. Furthermore, personal opinions may not have been expressed clearly, which may affect the scope of the research analysis. Fourth, this study was conducted as an elective course, which has restrictions on the number of students permitted to enroll. Therefore, a large participant population was not possible, which reduces the potential to extrapolate the research findings. Finally, undergraduate nursing students were enrolled, and it is inferred that the effectiveness of other educational systems (such as specialist departments, master students) or medical students is unknown. Further, only qualitative research methods were used. To further explore the efficacy of INL, a quantitative research method would be required to strengthen inference and generalizability.

## 5. Conclusions

This study strongly suggests that the use of the teaching strategy of intergenerational narrative learning to explore students’ learning experience in the course of older adults with chronic diseases led to improved understanding of the personal experience and the core beliefs of patients in their pre-present and future chronic disease management states. Through in-depth dialogue with these patients, students gained practical experience and knowledge from the patients’ stories. Intergenerational narrative learning stimulates students’ interest in learning and enhances their caring attitude toward chronically ill patients.

## Figures and Tables

**Figure 1 healthcare-09-01291-f001:**
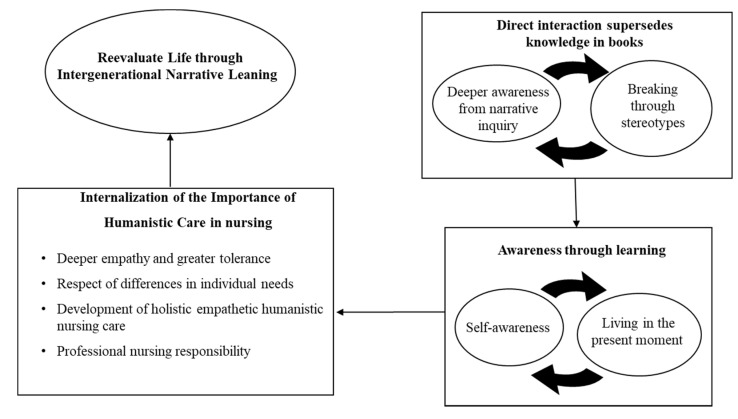
Structure of inspiration toward improved humanistic caring through intergenerational narrative learning (INL).

**Table 1 healthcare-09-01291-t001:** Demographic characteristics for nursing students.

Group	n	Sex	Age, Years (Range)
A	7	7 Female	20–21
B	7	7 Female	20–22
C	7	6 Female 1 Male	20–21
D	7	6 Female 1 Male	20–22
E	7	7 Female	20–21
Total/ range	35	33 Female 2 Male	20–22

## Data Availability

The data that support the findings of this study are available on request from the corresponding author, M.-Y.Y. The data are not publicly available due to their containing information that could compromise the privacy of research participants.

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
