# Peer review of "Intergenerational Narrative Learning to Bridge the Generation Gap in Humanistic Care Nursing Education"

_healthcare, 2021, doi:10.3390/healthcare9101291_

Round 1
Reviewer 1 Report
Thank you for the opportunity to review this article. The title of your manuscript and the opening paragraph do not seem to be part of the same argument. I found the first two sentences to be erroneous and misleading as well as the references for third sentence which refers to medical students and not nursing students. In summary, the introduction and background does not acknowledge the breadth of literature on caring and humanism in nursing education, research and practice. There is no clear link to narrative approach to narrative inquiry to storytelling...these are not interchangeable. You then move to intergenerational learning again there is a plethora of nursing literature on this subject. I suggest you do a comprehensive literature review that supports your study and clearly highlights what your study offers to current evidence and knowledge discourse.
You are interchanging the terms intergenerational learning and intergenerational narrative learning. Are they the same thing? A clear description of the concept being studied is necessary. It looks like you are naming an approach INL and that ought to be clearly described and threaded throughout the analysis and conclusion.
The learning activity shows great promise by having students have a guided discussion with a 'real' person and then reflect on what they learned. Your analysis was well done and the results have a lot to offer nurse educators about a learning activity that served to shift student nurses thinking about working with elderly people who have chronic illness.
Author Response
- The title of your manuscript and the opening paragraph do not seem to be part of the same argument. I found the first two sentences to be erroneous and misleading as well as the references for third sentence which refers to medical students and not nursing students.
Response 1
Thank you very much for your comments. The title and the opening paragraph have been revised. I have rechecked the references and made revision. Please refer to line 1-3 on the first paragraph of the introduction section (Page 1).
- In summary, the introduction and background does not acknowledge the breadth of literature on caring and humanism in nursing education, research and practice
Response 2
Thank you very much for your comments. The related descriptions have been added in the line 4-7 on the first paragraph of introduction section (Page 1). The structure of the introduction section also has been reorganized to highlight related issues.
- There is no clear link to narrative approach to narrative inquiry to storytelling...these are not interchangeable. You then move to intergenerational learning again there is a plethora of nursing literature on this subject. I suggest you do a comprehensive literature review that supports your study and clearly highlights what your study offers to current evidence and knowledge discourse.
Response 3
Thank you very much for your comments. We have changed “narrative inquiry” as ‘’narrative approach’’ because the current study applied the narrative approach in intergenerational learning. As your suggestion, we have reviewed the literature and highlighted what the current study offers to current evidence and knowledge discourse. Please refer to the red parts in the third and fourth paragraphs of the introduction section (Page 2).
- You are interchanging the terms intergenerational learning and intergenerational narrative learning. Are they the same thing? A clear description of the concept being studied is necessary.
Response 4
Thank you very much for your comments. Yes, they are the same thing. We have changed “intergenerational learning” as ‘’intergenerational narrative learning’’ for consistency.

Reviewer 2 Report
The manuscript is well written which states in a clear way and the storytelling process is vivid. However, since only qualitative study are adopted in this study, there exist the following limitations.
1. It is hard to reach some substantial conclusions due to the research scope (limited samples and time).
2. It is not sure to guarantee the obtained results are accordance with other sceneries. That is to say, the current research lacks the generalization to some extent.
Author Response
The manuscript is well written which states in a clear way and the storytelling process is vivid. However, since only qualitative study are adopted in this study, there exist the following limitations.
- It is hard to reach some substantial conclusions due to the research scope (limited samples and time).
Response 1
Thank you very much for your comments. This is indeed a limitation due to the limited sample and time. It has been written in the paragraph of limitation (line 9 -13 of the first paragraph on Page 9). We will also consider and improve this issue in our next study.
- It is not sure to guarantee the obtained results are accordance with other sceneries. That is to say, the current research lacks the generalization to some extent.
Response 2
Thank you very much for your comments. It has been added and explained in the final paragraph of limitation section (line 9 -13 of the first paragraph on Page 9).
